# Operator Learning Using Weak Supervision from Walk-on-Spheres

## Abstract

Training neural PDE solvers is often bottlenecked by expensive data generation or unstable physics-informed neural network (PINN) that involves challenging optimization landscapes due to higher-order derivatives. To tackle this issue, we propose an alternative approach using Monte Carlo approaches to estimate the solution to the PDE as a stochastic process for weak supervision during training. Recently, an efficient discretization-free Monte-Carlo algorithm called Walk-on-Spheres (WoS) has been popularized for solving PDEs using random walks. Leveraging this, we introduce a learning scheme called *Walk-on-Spheres Neural Operator (WoS-NO)* which uses weak supervision from WoS to train any given neural operator. The central principle of our method is to amortize the cost of Monte Carlo walks across the distribution of PDE instances. Our method leverages stochastic representations using the WoS algorithm to generate cheap, noisy, yet unbiased estimates of the PDE solution during training. This is formulated into a data-free physics-informed objective where a neural operator is trained to regress against these weak supervisions. Leveraging the unbiased nature of these estimates, the operator learns a generalized solution map for an entire family of PDEs. This strategy results in a mesh-free framework that operates without expensive pre-computed datasets, avoids the need for computing higher-order derivatives for loss functions that are memory-intensive and unstable, and demonstrates zero-shot generalization to novel PDE parameters and domains. Experiments show that for the same number of training steps, our method exhibits up to $8.75\times$ improvement in $L_2$-error compared to standard physics-informed training schemes, up to $6.31\times$ improvement in training speed, and reductions of up to $2.97\times$ in GPU memory consumption.

## 1 Introduction

Partial Differential Equations (PDEs) are fundamental mathematical tools for modeling a wide range of physical, geometric, and engineering systems. These equations describe how quantities vary across a domain, with applications spanning diverse scientific fields. A common thread across these fields is the challenge of solving the governing PDEs, a task often complicated by the complex and irregular geometries of the underlying domains.

Traditional methods for solving PDEs on complex geometries often fall into two broad categories: grid-based methods and grid-free approaches. Grid-based methods, such as finite difference method (FDM), finite volume method (FVM), and finite element method (FEM), discretize the domain into a grid or mesh and numerically solve PDEs over those partitions. Although these methods are highly accurate and widely used, their performance depends on the level of discretization, and they can become computationally prohibitive for large-scale problems or domains with complex geometries. Moreover, when the meshes are highly irregular or faulty, with cracks and sliver faces, they require computationally expensive geometric healing before they can be used in FEM (Chong et al., 2007).

Among the class of grid-free numerical techniques are the Monte Carlo-based methods (Sanz-Alonso & Al-Ghattas, 2024). These methods express the solution of the PDE as the recursive path integral formulation and define a Monte Carlo estimator for the integral equation (Li et al., 2024a). Among them, WoS is a notable technique that utilizes Brownian motion to estimate solutions to elliptic PDEs, notably the Poisson family of equations (Muller, 1956) by simulating particle trajectories and their interaction with domain boundaries. WoS avoids the complexities associated with grid dependencies by computing point-wise estimates, independent of other points. While conventional solvers like

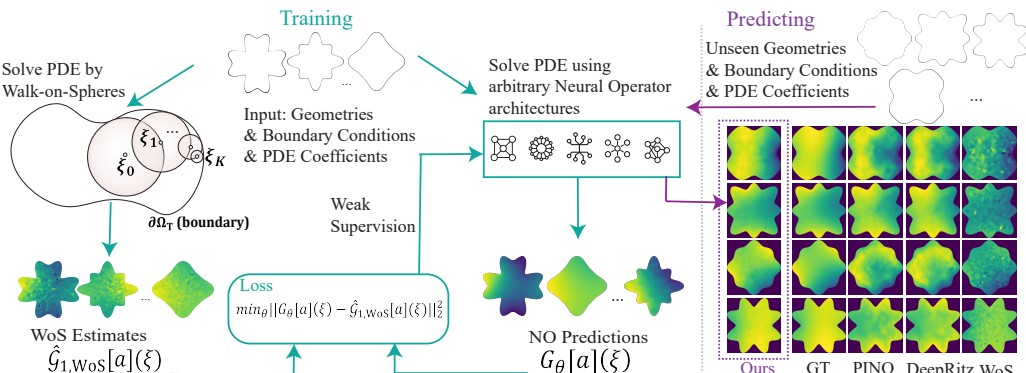

Figure 1: **Weak Supervision Loss with WoS:** Our algorithm learns the given family of parametrized Poisson equations $\Delta u = f$ on $\Omega_T \subset \mathbb{R}^d$ and $u|_{\partial\Omega_T} = g$ and is agnostic to underlying neural operator architecture. The WoS method defines the recursive process of the random walk, stopping once the boundary or the maximum number of steps is reached. The source contribution $f(\xi_i)$ is computed for each intermediate point $\xi_i$ before jumping to the next point $\xi_{i+1}$. We achieve variance reduction by controlling the number of walks $L$ to improve the fidelity of the weak solution. $\hat{\mathcal{G}}_{1,\text{WoS}}[a](\xi)$ denotes the estimation of 1-trajectory WoS estimation. $\xi_K$ denotes the termination condition where the boundary value $g(\xi_K)$ is added if the point is within the tolerance region and operator estimates $\mathcal{G}_\theta[a](\xi_K)$ is used otherwise. We illustrate the overall learning process, with WoS integral serving as the weak supervision for the neural operator.

Finite Element Method (FEM) for Poisson equations are more efficient on simple domains, they encounter significant bottlenecks on complex, non-watertight geometries Sawhney & Crane (2020) where mesh generation, a prerequisite for FEM, is computationally expensive or prone to failure. By leveraging WoS, we bypass this volumetric meshing bottleneck entirely, enabling scalable learning directly on raw geometry. However, Monte Carlo methods such as WoS suffer from slow convergence due to the high variance (Sawhney & Crane, 2020). Achieving accurate estimates requires performing a large number of particle walks to reach the domain boundary, often of the order of $10^4$ to $10^6$, to converge to the expected solution, leading to significant computational overhead.

Recent works have explored the integrations of neural networks with WoS to accelerate PDE solvers (Li et al., 2023; Nam et al., 2024). These methods improve the solution accuracy, reduce the prediction variance, and amortize the prediction cost by using neural networks to learn from rough estimations from WoS, reducing the computational cost of explicitly simulating numerous random walks. However, such approaches are often parameterization-dependent: changes in the PDE parameters (e.g., source terms or coefficients) or variations in the domain geometry require the network to be retrained from scratch. This limitation hinders their ability to support zero-shot generalization on complex problems.

To this end, neural operators (Li et al., 2024b; Alkin et al., 2024; Li et al., 2025) offer a promising solution by directly learning the mapping between PDE coefficient and boundary functions to the solution, enabling zero-shot generalization to unseen geometries and PDEs. Such data-driven neural operator frameworks, however, rely on ground-truth solutions typically generated using FEM a priori. While these models achieve zero-shot generalization, the necessity of pre-computed training data introduces significant computational and memory overhead, particularly for large-scale meshes and high-dimensional problems. Extensions of physics-informed neural networks (PINNs) to neural operators Li et al. (2024c) aim to overcome the need for data by minimizing the PDE residual; however, this requires (higher-order) derivative computations and suffers from optimization challenges (Krishnapriyan et al., 2021), particularly when the training data are scarce.

**Our Approach:** We enable physics-informed training of neural operators without the need for precomputed data by using pointwise Walk-on-Spheres estimates. In particular, we define a derivative-free regression objective using weak supervision from solution estimates generated with a minimal number of WoS trajectories. Due to the unbiasedness of the estimates, this simple and inexpensive regression objective still learns the groundtruth solution operator. Moreover, our strategy amortizes the cost of WoS simulations across the family of PDEs, enabling the trained operator to infer solutions for unseen PDE instances and geometries in fractions of a second. While prior (neural) WoS approaches incur significant costs to adapt to new problem instances, our operator learning framework achieves an inference cost of $\mathcal{O}(1)$ for unseen configurations. We demonstrate strong generalization, achieving

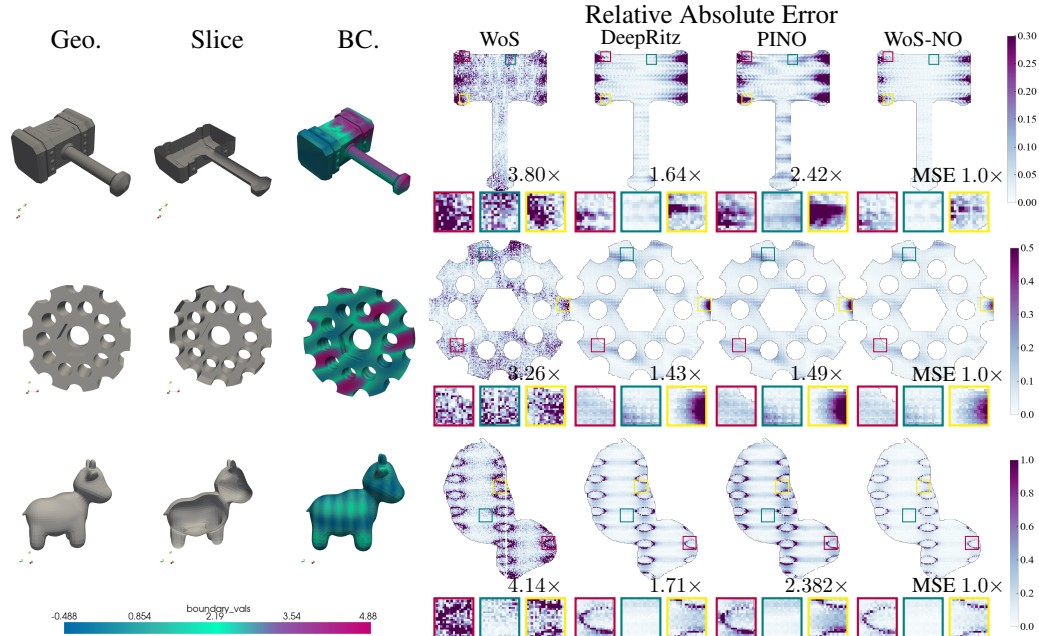

Figure 2: Given arbitrary, unseen input geometry and boundary conditions, we compare our method (WoS-NO) with WoS at equal execution time, and with DeepRitz and PINO at equal training time. We visualize the relative absolute error against the analytic solution (the ground truth). We show that WoS-NO achieves the strongest performance (lowest relative absolute error) in comparison with other baselines. During equal-time training, WoS-NO achieves 2.1× overall improvement than PINO and 1.59× than DeepRitz. During inference, WoS-NO achieves 3.73× better performance than WoS under the same time constraint.

8.75× improvement in comparison with physics-informed learning while reducing the memory by 2.4×. Furthermore, we illustrate the zero-shot generalization of our method for surface Laplace inpainting and fluid simulations.

**Problem scope**    We focus our analysis on the family of linear elliptic PDEs, specifically Poisson and Laplace equations. While the Walk-on-Spheres (WoS) algorithm is classically rooted in this domain, limiting the scope to the Poisson family is driven by two factors: geometric robustness and computational ubiquity. Solving Poisson-type equations is a computational bottleneck in several engineering and biological applications. Stochastic methods for these equations have been successfully deployed in molecular dynamics Mascagni & Simonov (2004), electrostatics for integrated circuit design Le Coz & Iverson (1992); Hwang et al. (2010); Rodriguez et al. (2025), and flow in porous media Hwang et al. (2000). Furthermore, recent advances leverage similar harmonic field formulations for autonomous robotic exploration in unknown environments Kotsinis et al. (2025), robust biharmonic skinning in computer graphics Dodik et al. (2024), and modeling protein drift-diffusion dynamics in curved cell membranes Tran et al. (2022); Miller et al. (2025).

Moreover, while we utilize the Poisson formulation to establish the paradigm of weakly-supervised neural operators, the underlying stochastic framework can be extended to broader PDE families. The Monte Carlo methodology has already been adapted to solve transient diffusion problems Shalimova & Sabelfeld (2024), reaction-diffusion equations Sabelfeld (2024), the Lamé equation for elasticity Aksyuk et al. (2023), and electron transport Kablukova et al. (2023). More recently, these methods have extended to non-linear advection-dominated regimes, including the Navier-Stokes and Burgers equations Sabelfeld & Bukhasheev (2022), as well as turbulent vector field representation Kim & Lee (2025). By validating our neural operator framework on the foundational Poisson family, we provide the necessary groundwork for a learning paradigm that can eventually extend to these advanced stochastic estimators as they mature.

We empirically validate the advantages of using Monte-Carlo methods like WoS by training on non-watertight meshes from the ShapeNet dataset, a regime where traditional solvers typically require extensive manual repair, as shown in Figure 2. Furthermore, the Poisson equation serves as a critical computational primitive within broader non-linear physics, most notably as the pressure projection

step in incompressible fluid dynamics Jain et al. (2024). Therefore, validating our framework on this family provides a verifiable pathway toward accelerating the internal loops of higher-order non-linear solvers, which we discuss in Section 6.4.

Our key contributions can be summarized as follows.

**Weak Supervision and Data-Free Operator Training:** We introduce a learning paradigm that requires no pre-computed solution data from expensive solvers like FEM. Our method bypasses the optimization challenges and computational costs of physics-informed losses by instead regressing against weak supervision from cheap and unbiased estimations of the stochastic PDE solvers, such as WoS algorithm. We prove that such a paradigm shift from physics loss to stochastic loss reduces the GPU memory significantly while achieving stronger generalization and efficiency.

**Amortized Variance Reduction Across PDEs:** The proposed framework amortizes the cost of Monte Carlo walks across an entire distribution of PDE instances. Using weak supervision, the stochastic training process converges to the true solution operator, effectively learning to denoise the weak signals over the given family of Poisson PDEs.

**Zero-Shot Generalization:** The trained operator can predict new PDE instances (e.g., with different boundary values, coefficient functions, or geometries) in a single forward pass, without retraining or additional simulations.

## 2 BACKGROUND

We provide in this section necessary backgrounds on Walk-on-Spheres and neural operators for understanding WoS-NO.

### 2.1 MONTE CARLO METHODS FOR PDES

Grid-free Monte Carlo methods are particularly advantageous for problems involving complex geometries, high-dimensional spaces, or irregular domains where traditional grid-based numerical methods face significant meshing bottlenecks (Yilmazer et al., 2024; Sawhney et al., 2022; 2023; Yu et al., 2024). The foundation of Walk-on-Spheres (WoS) (Muller, 1956) lies in the probabilistic interpretation of PDEs, specifically connecting the solution of Poisson equations to the expected values of stochastic Brownian motion. Crucially, WoS computes the solution at any query point independently, relying solely on the statistical properties of random walks without requiring a global mesh or linear system solve. While WoS has been successfully applied in computer graphics (Sawhney & Crane, 2020) and electrostatics (Bossy et al., 2010), it relies on sampling efficiency. Recent approaches have introduced variance reduction techniques, such as gradient control variates (Li et al., 2024a; Sawhney & Crane, 2020) and boundary value caching (Miller et al., 2023), to improve convergence rates. In the context of deep learning, hybrid approaches have emerged that use neural networks for variance reduction (Nam et al., 2024) or as cached surrogates (Li et al., 2023). However, these methods largely focus on accelerating the WoS solver itself rather than using WoS as weak supervision for training models.

### 2.2 NEURAL OPERATORS

Neural operators define a class of discretization-agnostic, data-driven solvers that learn mappings between infinite-dimensional function spaces (Azizzadenesheli et al., 2024; Kovachki et al., 2023; Lu et al., 2021; Viswanath et al., 2023). Unlike traditional solvers that depend on fixed mesh resolutions, these methods learn integral kernel operators, allowing for zero-shot super-resolution. Despite their inference efficiency, standard neural operators are fundamentally data-driven. To reduce data dependence, Physics-Informed Neural Operators (PINO) integrate PDE constraints directly into the loss function (Li et al., 2024c). However, PINO reintroduces significant optimization challenges: the loss landscape becomes highly complex, leading to training instabilities and sensitivity to hyperparameter tuning (Wang et al., 2021; Lin et al., 2025). Furthermore, PINO relies on automatic differentiation to compute PDE residuals, which incurs high memory costs and scales poorly with geometric complexity (Nam et al., 2024). While recent approaches like the Multi-Level Monte Carlo Operator (MLMC) attempt to accelerate training by decomposing the loss across fidelity levels (Rowbottom et al., 2025), they still fundamentally rely on the existence of pre-computed ground truth data. This creates a clear need for a training paradigm that is both data-free (like PINO) and computationally stable, a gap our WoS-NO framework addresses.

## 3 PROBLEM SETTING

Let $\Omega_T \subset \mathbb{R}^d$ be an open, bounded, connected, and sufficiently regular domain with $T \in \mathcal{T}$ as our distance function (e.g. signed distance function) defining the shape of the geometry. We consider the Poisson family of equations with source function family $\mathcal{F}$, boundary function family $\mathcal{B}$, and solution function family $\mathcal{U}$. The parameterized elliptic PDE problem with Dirichlet boundary conditions is given by the following system:

$$\begin{cases} \mathcal{P}[u] = f, & \text{on} \quad \Omega_T, \\ u = g, & \text{on} \quad \partial\Omega_T, \end{cases} \tag{1}$$

with a differential operator

$$\mathcal{P}[u] := \tfrac{1}{2}\text{Tr}(\sigma\sigma^\top \text{Hess}_u) + \mu \cdot \nabla u. \tag{2}$$

where $f \in \mathcal{F}$ is source function and $g \in \mathcal{B}$ boundary function both defined on the domain $\Omega_T$ and assumed to be sufficiently smooth.

Assuming the solution operator of the elliptic equation exists in the parameterized PDE family, our goal is to learn a solution operator $\mathcal{G} : \mathcal{A} = \mathcal{T} \times \mathcal{F} \times \mathcal{B} \to \mathcal{U}$, such that $a = (T, f, g) \mapsto u$, where $u$ is the solution of the equation 1 over arbitrary combinations of $T$, $f$ and $g$.[1]

### 3.1 WALK-ON-SPHERES FROM THE OPERATOR PERSPECTIVE

We generalize the derivation from neural Walk-on-Spheres (Nam et al., 2024) from the operator perspective. For our paper, we focus on Poisson equations. Poisson equations can be reformulated as stochastic differential equations driven by a random variable $\xi$, defined over the domain $\Omega_T$. The stochastic differential equation is given by $\mathrm{d}X_t^\xi = \mu(X_t^\xi)\mathrm{d}t + \sigma(X_t^\xi)\mathrm{d}W_t$, $X_0^\xi \sim \xi$, where $W_t$ is a standard $d$-dimensional Wiener process, and $\mu$ and $\sigma$ are time-dependent parameters with $\mu = 0$ and $\sigma = \sqrt{2}I$ for Poisson equations.

By utilizing Walk-on-Spheres, we can reformulate the solution operator $\mathcal{G}_{\text{WoS}} : \mathcal{A} \to \mathcal{U}$ for the Poisson equation as the expectation of the following form:

$$\mathcal{G}[a](\xi) = \mathcal{G}_{\text{WoS}}[a](\xi) = \mathbb{E}[g(X_\tau^\xi) - \sum_{k \geq 0} \int_0^{\tau_k} f(X_t^\xi)\mathrm{d}t | \xi], \tag{3}$$

where we define $\tau_k = \tau(B_{r_k}, \xi_k) \triangleq \inf\{t \in [0, \infty) : X_t^{\xi_k} \notin B_{r_k}\}$ to be the stopping time within a sphere $B$ of radius $r_k$ centered at $\xi_k$ and $r_k = \text{dist}(\xi_k, \partial\Omega_T) \in (0, \infty)$ at the $k$-th step of the random walk, $\tau = \tau(\Omega, \xi)$, and $\xi_{k+1} \sim X_{\tau_k}^{\xi_k} \sim \mathcal{U}(\partial B_{r_k}(\xi_k)), \xi_0 = \xi$. We defined more detailed derivations of each term in Appendix B.

In practice, we can approximate the operator with Monte Carlo (MC) simulation to obtain $\mathcal{G}_{\text{WoS}}[a](\xi) \approx \hat{\mathcal{G}}_{L,\text{WoS}} = \frac{1}{L}\sum_{i=1}^L \text{WoS}^i[a](\xi)$, where $L$ is the number of trajectories with

$$\text{WoS}^i[a](\xi) = g(\xi_K^i) - \sum_{k=0}^{K-1} |B_{r_k^i}(\xi_k^i)| f(\gamma_k^i) G_{r_k^i}(\gamma_k^i, \xi_k^i), \tag{4}$$

and $K$ is the maximum step of WoS, $a = (T, f, g) \in \mathcal{A}$, $G$ is the Green's function defined in Appendix B, and $\gamma_k^i \sim \mathcal{U}(B_{r_k^i}(\xi_k^i))$.

Under the assumption that the solution exists as shown in (Le Gall, 2016), we have the equation $\mathcal{G}[a](\xi) = \mathcal{G}_{\text{WoS}}[a](\xi) \approx \hat{\mathcal{G}}_{L,\text{WoS}}[a](\xi)$, meaning that we can approximate the ground truth solution $\mathcal{G}$ with stochastic estimator $\hat{\mathcal{G}}_{L,\text{WoS}}$ while controlling the fidelity with the number of trajectories $L$.

### 3.2 WALK-ON-SPHERES AS WEAK SUPERVISION

We propose the following general loss formulation to learn families of parameterized Poisson equations as a conditional expectation

$$\mathcal{L}_\theta = \mathbb{E}[\|\mathcal{G}_\theta - \mathcal{G}\|^2] = \mathbb{E}[\mathbb{E}[\|\mathcal{G}_\theta[a](\xi) - \mathcal{G}_{\text{WoS}}[a](\xi)\|^2|\xi]], \tag{5}$$

---

[1]Whenever the context is clear, we write $\Omega_T \triangleq \Omega$ for simplicity

which follows from the tower property and the definition of the WoS operator.

We generate *unbiased high-variance* weak supervision with WoS over a small number of trajectories $L \leq 10$. This allows fast ground truth generation on the fly in each epoch for regressing the neural operator. The empirical loss then becomes

$$\hat{\mathcal{L}}_\theta = \frac{1}{NM} \sum_{j=1}^{M} \sum_{i=1}^{N} \|\mathcal{G}_\theta[a^j](\xi^i) - \hat{\mathcal{G}}_{L,\text{WoS}}[a^j](\xi^i)\|^2. \tag{6}$$

The entire pipeline is shown in more visual details in Figure 1. Moreover, the cache incurs an additional space complexity of $O(MN)$, where $M$ is the number of PDE instances and $N$ is the number of points per instance. We add ablation studies on design choices and hyperparameters in Appendix F.

## 4 SCENE SETUP

This section outlines the setups for our experiments. We structure our evaluation across two fundamental problem classes: linear Poisson with constant and varying coefficients. The WoS framework is most directly suitable for linear and screened forms of Poisson PDEs. In order to extend WoS to varying coefficients test cases, we linearized the governing equation into a screened form using the delta-tracking method (Sawhney et al., 2022) and formulated the delta-tracking variant of WoS loss for the operator. Our operator thereby takes both spatial coefficient functions and geometry as inputs.

### 4.1 POISSON EQUATIONS ON PARAMETERIZED DOMAINS

We follow the settings proposed in (Qin et al., 2022) to define a linear version of the Poisson equation on parameterized geometries as follows

$$\begin{aligned} \Delta u(x) &= f(x), & x \in \Omega_T \\ u(x) &= g(x), & x \in \partial\Omega_T \end{aligned} \tag{7}$$

where $u \in \mathcal{U}$ and $\Omega_T \subset \mathbb{R}^2$. The source function is a sum of radial basis functions denoted by $f(x) = \sum_{i=1}^{2} \beta_i e^{\|x-\mu_i\|_2^2}$, where $\beta \in \mathbb{R}^1$ and $\mu_i \in \mathbb{R}^2$. The boundary term is a periodic function defined in the polar-coordinate system as follows $g(x) = b_0 + b_1 cos(\theta) + b_2 sin(\theta) + b_3 cos(2\theta) + b_4 sin(2\theta)$, where $b_{0:4} \sim \mathcal{U}(-1, 1)$.

### 4.2 SECOND ORDER PDEs WITH SPATIALLY VARYING COEFFICIENTS

We consider PDEs with spatially varying coefficients defined over irregular domains. The second-order elliptic PDE is denoted as

$$\begin{aligned} \nabla \cdot (\alpha \nabla u(x)) + \vec{\omega} \cdot \nabla u(x) - \sigma u(x) &= -f(x), & x \in \Omega_T \\ u(x) &= g(x), & x \in \partial\Omega_T \end{aligned} \tag{8}$$

where, $\Omega_T \subset \mathbb{R}^3$ is the domain defined by distance $T$, $\alpha \in C^2(\Omega_T, \mathbb{R}_+)$, $\vec{\omega} \in C^2(\Omega_T, \mathbb{R}^3)$, and $\sigma \in C(\Omega_T, \mathbb{R}_{>0})$. As presented in (Li et al., 2023; Sawhney et al., 2022), $\alpha$ represents the *diffusion coefficient*, $\vec{\omega}$ is the *drift coefficient* and $\sigma$ denotes the *absorption coefficient*.

We consider a variant of this PDE by assuming $\vec{\omega} = 0$, and define a delta-tracking-based WoS loss. This is done so by making the following substitutions to equation 8. $U(x) = \sqrt{\alpha(x)}u(x)$, $g'(x) = \sqrt{\alpha(x)}g(x)$, $f'(x) = \frac{\sqrt{\alpha(x)}}{\alpha(x)}f(x)$, $\sigma'(x) = \frac{\sigma(x)}{\alpha(x)} + \frac{1}{2}(\frac{\Delta\alpha(x)}{\alpha(x)} + \frac{|\nabla \ln(\alpha(x))|^2}{2})$. This reformulation leads to a screened variant of equation 3:

$$\mathcal{G}_{\text{WoS},\Delta}[a](\xi) = \mathbb{E}[e^{-\bar{\sigma}\tau_k} g'(X_{\tau_k}^\xi) + \sum_{k \geq 0} \int_0^{\tau_k} f(X_t^\xi, U)dt | \xi], \tag{9}$$

We derive the delta-tracking algorithm in Appendix C. This is adapted to a new loss function as follows:

$$\hat{\mathcal{L}}_{\theta,\Delta} = \frac{1}{NM} \sum_{j=1}^{M} \sum_{i=1}^{N} \| \mathcal{G}_\theta[a^j, \alpha^i, \sigma^i](\xi^i) - \hat{\mathcal{G}}_{L,\text{WoS},\Delta}[a^j](\xi^i) \|^2, \tag{10}$$

where $\hat{\mathcal{G}}_{L,\text{WoS},\Delta}[a^j](\xi^i)$ is an L-trajectory empirical mean of estimations with Delta tracking such that $\hat{\mathcal{G}}_{L,\text{WoS},\Delta}[a](\xi) \approx \mathcal{G}_{\text{WoS},\Delta}[a](\xi)$.

We provide an abstract view of the training pipeline of WoS-NO in Algorithm 1 and the definition of diffusion and absorption terms in Appendix C.

## 5 IMPLEMENTATION

The WoS framework is implemented using the `zombie` C++ library (Sawhney & Crane, 2020). We leverage python bindings to make calls from the training framework. While we do not generate datasets for training, we generate domains (shapes) and PDE family coefficients. We provide the data generation details along with training details in Appendix D.

## 6 EXPERIMENTS

This section presents the empirical validation of our proposed framework. Our evaluation is structured into three parts: benchmarking against baseline methods, quantifying amortization gains, and demonstrating generalization to diverse applications.

First, we benchmark our framework against two classes of data-free PDE solvers: Physics-Informed Neural Operators (PINOs) and Deep Ritz methods (which we extend to neural operators from Yu et al. (2018)). For the comparison against the traditional Walk-on-Spheres (WoS) solver, we conduct an equal sample analysis and equal time analysis. This involves averaging the outputs from multiple independent runs of both our trained operator and the WoS solver to evaluate and compare the consistency of their solutions under an identical sampling budget.

Second, we quantify the amortization gains of our operator, which is the computational speedup at inference time that offsets the initial cost of training. We measure this in two settings. For unseen PDE parameterizations, we quantify the gain by measuring the throughput of the traditional WoS solver; specifically, we calculate the number of new problem instances it can solve from scratch within the total wall-clock time required for our operator to converge to an L2 relative error of $10^{-3}$. To analyze scalability, we then compare the wall-clock time of our neural solver against the WoS solver across various resolutions, demonstrating our method's efficiency on finer discretizations.

Finally, we showcase the versatility of our framework by applying it to a diverse set of domains beyond physics, including fluid dynamics and image inpainting.

### 6.1 COMPARISONS WITH BASELINES

We benchmark our proposed WoS-NO against two established data-free baselines: the Physics-Informed Neural Operator (PINO) and the Deep Ritz Operator. We evaluate all methods on key metrics including training time, final accuracy on unseen PDEs, and computational resource consumption. The results are summarized in Table 1 and Table 2.

Table 1: Performance and compute comparison of our WoS-NO against data-free baselines on the linear Poisson family. All methods were trained for 20,000 steps, with the evaluation metrics averaged over 1000 unseen linear parameterizations. Our approach achieves the highest accuracy with significantly lower GPU resource consumption.

| APPROACH | TRAINING TIME (Min) (20K steps) | AVERAGE L2 ERROR $\pm$Std | PEAK GPU MEMORY (MB) | PEAK GPU POWER USAGE (W) |
|---|---|---|---|---|
| PINO | 85.25 | $2.5e^{-3} \pm 4.7e^{-3}$ | 1523.5 | 78.43 |
| Deepritz Operator | 35.401 | $1.0e^{-2} \pm 1.6e^{-2}$ | 859.43 | 49.52 |
| WoS-NO | **13.5** | $\mathbf{8.2e^{-4} \pm 5.4e^{-4}}$ | **627** | **48.81** |

The data in Table 1 and Table 2 highlight the advantages of our method. Our WoS-NO achieves the lowest $L_2$ error while simultaneously reducing peak GPU memory and power consumption compared to both baselines. It remains faster than both PINO and Deepritz.

Table 2: Performance and compute comparison of our WoS-NO against data-free baselines on Poisson equations with spatially varying coefficients. All methods were trained for 50,000 steps, with the evaluation metrics averaged over 1000 unseen spatially varying parameterizations. Our approach achieves the highest accuracy with significantly lower GPU resource consumption.

| APPROACH | TRAINING TIME (Min) (50K steps) | AVERAGE L2 ERROR $\pm$Std | PEAK GPU MEMORY (MB) | PEAK GPU POWER USAGE (W) |
|---|---|---|---|---|
| PINO | 411 | $9.4e^{-3} \pm 1.1e^{-2}$ | 8587.9 | 122.85 |
| Deepritz Operator | 198 | $1.1e^{-2} \pm 2.4e^{-2}$ | 3903.1 | 116.59 |
| WoS-NO | **188** | $\mathbf{9.0e^{-3} \pm 7.9e^{-3}}$ | **2886.3** | **103.90** |

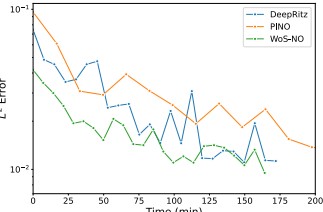 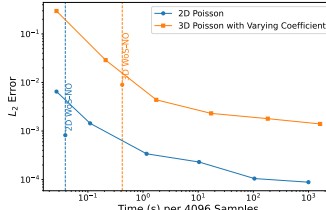 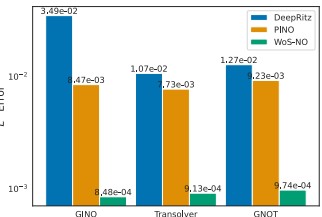

Figure 3: **Left:** Training Poisson equations with spatially varying coefficients with equal time for 200 minutes, WoS-NO demonstrates the lowest $L_2$ error while converging the fastest. In contrast, PINO requires a much longer time to converge. **Middle:** After WoS-NO training is finished, we compare the amount of time needed to achieve the same level of accuracy as $L_2$-error to achieve the same $L_2$ error as a well-trained WoS-NO for 4096 pointwise estimations. **Right:** GINO, Transolver and GNOT are trained on DeepRitz, PINO and WoS-NO losses, and across all three operator architectures, WoS-NO is the strongest with the lowest $L_2$-error.

In Figure 1 right, we make comparisons of predictions for the linear Poisson family over diverse geometries using WoS-NO, ground truth, PINO, DeepRitz, and WoS. We see that WoS-NO's predictions are much closer to the ground truth than other methods. Moreover, while WoS results in very noisy predictions, WoS-NO smooths out noises from training and makes smooth predictions in different geometries, proving its efficiency and strong performance in learning the solution operator for parametrized PDEs with weak supervision.

### 6.2 AMORTIZATION

In order to make a fair comparison of performances between DeepRitz, PINO, and WoS-NO, we perform an equal time training comparison of the three methods with Poisson equations with spatially varying coefficients. We fix the number of training time to 200 minutes and train three models with optimized hyperparameters. Figure 3 left shows that under the fixed time framework, WoS-NO achieves the lowest $L_2$-error ($9.0e^{-3}$) while converging fastest in comparison with both DeepRitz and PINO. Furthermore, we discover that PINO takes the longest to converge due to the computation of higher-order derivatives as well as its complex loss landscapes.

Furthermore, we compare for Poisson equations with spatially varying coefficients the amount of time it requires to achieve the same accuracy as a well-trained WoS-NO. As shown in Figure 3 middle, WoS requires roughly $7.2\times$ and $1.9\times$ longer inference time in comparison with WoS-NO to achieve the same level of $L_2$ error.

We further demonstrate the comparison of three architectures in inference on ShapeNet. Figure 2 shows that over 3 different geometries from ShapeNet, WoS-NO achieves the strongest zero-shot inference with $2.1\times$ overall improvement than PINO and $1.59\times$ than DeepRitz. Moreover, under the same time constraint, WoS-NO achieves $3.73\times$ better performance than WoS, demonstrating empirically that neural operators learn to smooth out noises in stochastic estimations.

### 6.3 ARCHITECTURE-AGNOSTIC LOSS

To demonstrate that WoS-NO is architecture-agnostic, we train GINO, Transolver and GNOT on linear Poisson equations with DeepRitz, PINO, and WoS-NO losses. As shown in Figure 3 right, regardless of which neural operator architecture we use, we see the lowest $L_2$ error with WoS-NO error in comparison with DeepRitz($100\times$) and PINO($10\times$). This shows that WoS-NO is a general framework that is applicable to any operator.

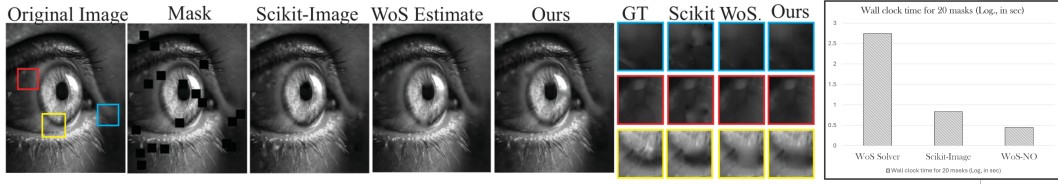

Figure 4: **Quantitative comparison for biharmonic inpainting.** We evaluate our pre-trained operator against a traditional Walk-on-Spheres (WoS) solver and the `scikit-image` baseline. The plot shows the final error versus total runtime (log scale, in seconds) required to inpaint 20 masks. The average MSE compared to the ground truth for our method is $2.8e^{-3} \pm 7.2e^{-3}$, WoS ($5.7e^{-3} \pm 5.2e^{-3}$) and `scikit-image` ($5.4e^{-3} \pm 5.0e^{-3}$). The total wall clock time for WoS solver is 557.2s, Scikit-Image 6.8s and WoS-NO 2.8s.

### 6.4 GENERALIZATION TO PHYSICAL AND GEOMETRIC DOMAINS

A key advantage of our framework is its capacity for zero-shot generalization. To demonstrate this, we apply our operator, pre-trained on parametrized Poisson problems, to solve tasks from typical 2D and 3D problems that leverage Poisson methods, without any retraining or fine-tuning. This is made possible by reformulating each target problem with stochastic integrals. We discuss three classes of problems - von Karman vortex problem, biharmonic inpainting, and Poisson surface reconstruction.

**Biharmonic inpainting:** We demonstrate the zero-shot generalization of our operator to higher-order PDEs by applying it to biharmonic image inpainting. The task is governed by the fourth-order equation $\Delta^2 u = 0$ within the masked image domain $D$. We reformulate this problem by introducing an intermediate function $v = \Delta u$, which decomposes the single fourth-order PDE into a system of two coupled second-order equations:

$$\begin{cases} \Delta v = 0, & \text{with } v|_{\partial D} = f \\ \Delta u = v, & \text{with } u|_{\partial D} = g \end{cases} \tag{11}$$

This system is solved recursively: we first apply our pre-trained operator to solve the Laplace equation for $v$, then use the resulting solution $v$ as a source term to solve the Poisson equation for the final intensity $u$. For our evaluation, we inpaint 20 masks, of dimensions 150x150 in pixel-space, and report the average performance. As shown in Figure 4, our method achieves competitive accuracy while being orders of magnitude faster than the traditional WoS solver.

First, we apply our trained operator to solve the Laplace equation for $v$. Then, using the resulting solution $v$ as a source term, we apply our operator a second time to solve the Poisson equation for the final image intensity $u$. For this problem, we consider a mask size of 150x150 pixels, and generate 20 masks, our results are the average over those. We present the results in Figure 4.

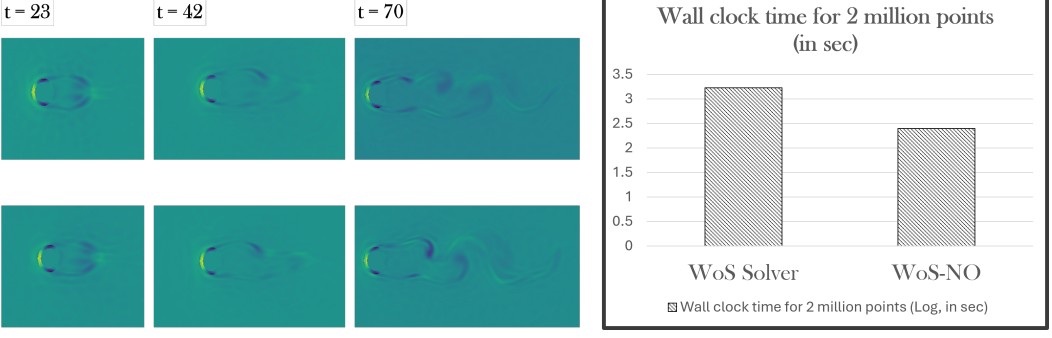

Figure 5: Qualitative and quantitative results showing the pressure field magnitude in the von Kármán vortex. (**Left**) Visual comparison of the pressure fields computed by the baseline WoS solver Jain et al. (2024) (top) and our pre-trained operator (bottom). Our method achieves a relative error of $2.5 \times 10^{-1}$ with respect to the baseline. (**Right**) Log-scale plot of the total wall-clock time required to compute the pressure solution over 70 simulation time-steps, comparing the computational cost of both methods. The total wall clock time for WoS solver is 1698.2s and WoS-NO 251.1s

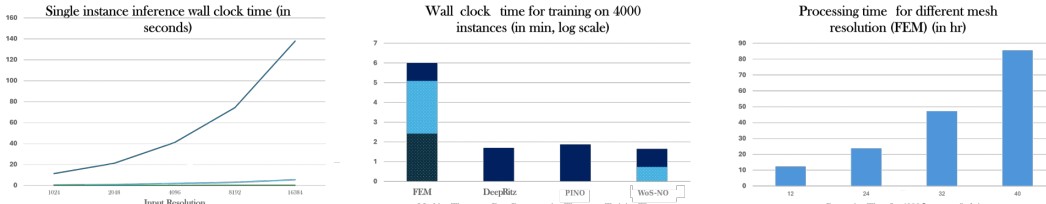

Figure 6: **Left:** This graph showcases the inference time of a single instance, as a function of domain size. In the case of FEM, it denotes the resolution of the mesh, while for the WoS-based methods, it's the number of query points **Middle:** This figure highlights the amortization achieved with our approach. Meshing time includes time needed to create meshes in the domain. Data preprocessing time includes the simulation time to create the regression target (FEM solution and WoS simulation). Our approach does not require meshing, and the total time for WoS-NO is lower than PINO while approximately equal to DeepRitz total time, making it an efficient operator for training. **Right:** This figure highlights how the data preprocessing time for FEM scales with the resolution of the inputs. Increasing the resolution results in a direct increase in computation time (hours). This is avoided by WoS-NO since WoS is highly parallelizable on GPU and does not depend on the input resolution.

**von Kármán Vortex:**   We investigate our operator's utility as a fast, approximate solver by applying it to the pressure projection step of a von Kármán vortex simulation. This presents a particularly challenging zero-shot generalization task, as the simulation requires solving a sequence of previously unseen, screened Poisson equations with Neumann boundary conditions—a different problem class from our training data.

As shown in Figure 5, our operator offers a significant computational speedup compared to the traditional WoS solver. The resulting pressure field, while having an $L_2$-relative error of $2.5e^{-1}$, is obtained orders of magnitude faster. This outcome highlights the operator as a fast, generalizable approximator. Its ability to produce an approximation at high speed paves the way for integrating it within hybrid methods, such as providing an early termination criterion for expensive Monte Carlo walks as proposed by Nam et al. (2024).

### 6.5 COMPARISON WITH FEM SOLVERS

To make further comparisons, we compare WoS-NO with FEM-based traditional solvers in terms of speed. As shown in Figure 6 left, we see that the FEM solver requires exponentially increasing inference time for inference in finer resolution. In contrast, while WoS cost increases slowly, WoS-NO has almost zero increase in time cost, demonstrating its scalability to high-fidelity PDEs.

In Figure 6 middle, we compare total training time, which includes data preprocessing time, meshing, and training time for FEM, DeepRitz, PINO and Wos-NO. FEM demonstrates the highest total time due to its requirement for meshing and processing for solutions. While WoS-NO requires extra time for preprocessing to perform random walks, the total training time is much smaller than other methods since it avoids computations of gradients over higher-order derivatives.

Figure 6 right demonstrates the processing time for FEM with an increase in mesh resolution. We see that as we increase the mesh resolution from 12 to 40 ( $4\times$), the data processing time increases $8\times$, making it not scalable to higher and finer resolution tasks. In contrast, WoS-NO's processing time uses random walks with WoS and is highly parallelizable with GPU. Moreover, it is a meshless method and is independent of the resolution grid. As a result, it is a much more scalable option for complex geometries with fine resolutions.

## 7 FUTURE WORK

A challenging test for the generalizability of a PDE solver is Poisson surface reconstruction. Current methods, such as physics-informed neural fields, often require per-scene optimization and struggle to generalize to new inputs without retraining. Our framework, however, can be applied in a zero-shot manner by reformulating the task as a Poisson problem. The standard approach seeks to find an indicator function $\chi$ of the surface's interior whose Laplacian equals the divergence of the input point cloud's normal field, $N$, i.e., $\Delta\chi = \nabla \cdot \mathbf{N}$. One promising direction is to model the surface reconstruction as a stochastic integral; such an approach, which we leave for future work, could pave the way for a *truly general-purpose, training-data-free foundational Poisson solver*.

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

## A  BACKGROUND: OPERATOR LEARNING

A neural operator (Kovachki et al., 2023) is a data-driven approximation of mappings between function spaces denoted as $\mathcal{G}_\theta : \mathcal{A} \to \mathcal{U}$ such that $\mathcal{G}_\theta \approx \mathcal{G}$, where $\mathcal{G}$ is defined in equation 3.

A neural operator is composed of pointwise and integral operators, which can be represented as the discretization of the kernel-integral transform, where the learnable kernel is parameterized by neural networks. In our work, we propose the use of neural operators to learn the integral presented in Equation (3). We define a neural operator based on GINO (Li et al., 2024b) to approximate the unknown Green's function by formulating it as a kernel integral as shown below

$$\mathcal{G}[a](\xi) = u(\xi; a) \coloneqq \int_{\Omega_T} \kappa_\theta(\xi, \gamma) a(\gamma) \mathrm{d}\gamma. \tag{12}$$

In the discrete setting, this is represented as a summation, denoted as

$$\mathcal{G}[a](\xi) \approx \sum_j \kappa_\theta(\xi, \gamma_j) a(\gamma_j). \tag{13}$$

We follow the model architecture presented in GINO (Li et al., 2024b). The input, represented as point clouds, is projected onto a uniform latent grid, and the input functions $a = (f, g, T)$ are integrated by the FNO. We use a second GNO to evaluate the integral over a desired set of validation points defined over the same spatial domain but may differ from the input point clouds.

## B    DERIVATION OF WALK-ON-SPHERES

We present a more detailed derivation of Walk-on-Spheres.

Given equation 3.1, we apply Itô's lemma to the process $u(X_t^\xi)$ and can reformulate the Poisson equation as

$$g(X_\tau^\xi) = u(\xi) + \int_0^\tau f(X_t^\xi)\mathrm{d}t + \sqrt{2}\int_0^\tau \nabla u(X_t^\xi)\mathrm{d}W_t, \tag{14}$$

where $\tau = \tau(\Omega, \xi) = \inf\{t \in [0, \infty) : X_t^\xi \notin \Omega\}$ is the stopping time or the first exit time of the stochastic process from the domain $\Omega$.

By writing the equation in an expectation form, we can get

$$u(x) = \mathbb{E}\left[g(X_\tau^\xi) - \int_0^\tau f(X_t^\xi)\mathrm{d}t \big| \xi = x\right], \tag{15}$$

since $\int_0^\tau \nabla u(X_t^\xi)dW_t$ has an expectation zero, see, e.g., Baldi (2017, Theorem 10.2).

By restricting the domain $\Omega$ to an open sub-domain $\Omega_0 \subset \Omega$ with $\xi_0 = \xi \in \Omega_0$ and $\tau_0 = \tau(\Omega_0, \xi)$, we can rewrite the expectation as

$$u(\xi) = \mathbb{E}\left[u(X_{\tau_0}^\xi) - \int_0^{\tau_0} f(X_t^{\xi_0})\mathrm{d}t \big| \xi\right]. \tag{16}$$

To recursively solve the equation, we further define the random variable $\xi_1 \sim X_{\tau_0}^\xi$ with sub-domain $\Omega_1 \subset \Omega$ containing $\xi_1$ and $\tau_1 = \tau(\Omega_1, \xi_1)$. We get

$$u(X_{\tau_0}^\xi) \sim u(\xi_1) = \mathbb{E}\left[u(X_{\tau_1}^{\xi_1}) - \int_0^{\tau_1} f(X_t^{\xi_1})\mathrm{d}t \big| \xi_1\right]. \tag{17}$$

As a result, we can define a recursive solution

$$u(\xi) = \mathbb{E}\left[g(X_\tau^\xi) - \sum_{k \geq 0}\int_0^{\tau_k} f(X_t^{\xi_k})\mathrm{d}t \bigg| \xi\right]. \tag{18}$$

The second term in 18 can be computed by using Green's function such that

$$\mathbb{E}[\int_0^{\tau_k} f(X_t^\xi)\mathrm{d}t] = \int_{B_{r_k}(\xi_k)} f(y)G(\xi_k, y)\mathrm{d}y \tag{19}$$

which can be solved through Monte Carlo integrations such that

$$\mathbb{E}[\int_0^{\tau_k} f(X_t^\xi)\mathrm{d}t] = |B_{r_k}(\xi_k)|\mathbb{E}[f(y)G_{r_k}(y, \xi_k)] \tag{20}$$

where $y \sim \mathcal{U}(B_{r_k}(\xi_k))$ and

$$G_r(y, z) := \begin{cases} \frac{1}{2\pi}\log\frac{r}{\|y-z\|}, & d = 2, \\ \frac{\Gamma(d/2-1)}{4\pi^{d/2}}\left(\|y-z\|^{2-d} - r^{2-d}\right), & d > 2, \end{cases} \tag{21}$$

## C    DERIVATION OF DELTA TRACKING

We follow the derivation presented in (Sawhney et al., 2022).

Let us define the PDE to be as follows

$$\begin{aligned} \nabla \cdot (\alpha(x)\nabla u) - \sigma u(x) &= -f(x), & x \in \Omega \\ u(x) &= g(x), & x \in \partial\Omega \end{aligned} \tag{22}$$

We expand the second order term and apply the identity $\nabla \ln(\alpha(x)) = \nabla\alpha(x)/\alpha(x)$

$$\Delta u(x) + \nabla \ln(\alpha(x)) \cdot \nabla u(x) - \frac{\sigma(x)}{\alpha(x)} u(x) = -\frac{f(x)}{\alpha(x)} \tag{23}$$

We eliminate the first order operator by applying Girsanov transformation.

$$\begin{aligned} \Delta U(x) - \sigma'(x)U(x) &= -f'(x), \quad x \in \Omega \\ U(x) &= g'(x), \quad x \in \partial\Omega \end{aligned} \tag{24}$$

Where, $U(x) = \sqrt{\alpha(x)}u(x)$, $g'(x) = \sqrt{\alpha(x)}g(x)$, $f'(x) = \frac{\sqrt{\alpha(x)}}{\alpha(x)}f(x)$, $\sigma'(x) = \frac{\sigma(x)}{\alpha(x)} + \frac{1}{2}(\frac{\Delta\alpha(x)}{\alpha(x)} + \frac{|\nabla \ln(\alpha(x))|^2}{2})$

We introduce a coefficient $\bar{\sigma} > 0$, to shift the heterogenity to a source term. The equation then becomes

$$\begin{aligned} \Delta U(x) - \bar{\sigma}U(x) &= -f'(x, U), \quad x \in \Omega \\ U(x) &= g'(x), \quad x \in \partial\Omega \end{aligned} \tag{25}$$

where $f'(x, U) = f'(x) + (\bar{\sigma} - \sigma'(x))U(x)$. Using the Feynman-Kac formulation, we derive

$$U(\xi) = \mathbb{E}[e^{-\bar{\sigma}\tau}g'(X_\tau^\xi) + \int_0^\tau e^{-\bar{\sigma}t}f'(X_t^\xi, U)\mathrm{d}t|\xi]. \tag{26}$$

Similar to the derivation of the original Walk-on-Spheres, we define the integral as

$$U(X_{\tau_0}^\xi) \sim u(\xi_1) = \mathbb{E}[e^{-\bar{\sigma}\tau_1}U(X_{\tau_1}^{\xi_1}) + \int_0^{\tau_1} e^{-\bar{\sigma}t}f'(X_t^{\xi_1}, U)\mathrm{d}t|\xi_1]. \tag{27}$$

By solving this recursively, we find that

$$U(\xi) = \mathbb{E}[(\Pi_{k \geq 0}e^{-\bar{\sigma}\tau_k})g'(X_\tau^\xi) + \sum_{k \geq 0}(\Pi_{i \leq k}e^{-\bar{\sigma}\tau_i})\int_0^{\tau_k} e^{-\bar{\sigma}t}f'(X_t^{\xi_k}, U)\mathrm{d}t|\xi]. \tag{28}$$

To evaluate the solution iteratively, we need to reformulate $\mathbb{E}[e^{-\bar{\sigma}\tau_k}U(X_{\tau_k}^{\xi_k})|\xi_k]$ and $\mathbb{E}[\int_0^{\tau_k} e^{-\bar{\sigma}\tau_k}U(X_t^{\xi_k})\mathrm{d}t|\xi_k]$.

The first term can be approximated using the Poisson kernel $P^{\bar{\sigma}}$

$$\mathbb{E}[e^{-\bar{\sigma}\tau_k}U(X_{\tau_k}^{\xi_k})|\xi_k] = \int_{\partial B_{r_k}(\xi_k)} U(z)P^{\bar{\sigma}}(z)\mathrm{d}z, \tag{29}$$

and the second term can be approximated using Green's function $G^{\bar{\sigma}}$

$$\mathbb{E}[\int_0^{\tau_k} e^{-\bar{\sigma}t}U(X_t^{\xi_k})\mathrm{d}t|\xi_k] = \int_{B_{r_k}(\xi_k)} f'(y, U)G^{\bar{\sigma}}(\xi_k, y)\mathrm{d}y, \tag{30}$$

As such, the new integral can be recursively represented as a solution operator $\mathcal{G}_\Delta$ of screened Poisson equations

$$\begin{aligned} \hat{\mathcal{G}}_\Delta[a](\xi) = \frac{1}{\sqrt{\alpha(\xi)}}(&\int_{B(c)} f(y)G^{\bar{\sigma}}(\xi, y)dy + \\ &\int_{\partial B(c)} \sqrt{\alpha(z)}\hat{\mathcal{G}}_\Delta[a](z)P^{\bar{\sigma}}(\xi, z)dz) \end{aligned} \tag{31}$$

where, $G^{\bar{\sigma}}$ is Green's function and $P^{\bar{\sigma}}$ is the Poisson kernel, given by the normal derivative of Green's function at the boundary. Refer (Sawhney et al., 2022) for more detailed derivations.

## C.1 DEFINITION OF TERMS FOR POISSON WITH SPATIALLY VARYING COEFFICIENTS

In this section, we define all terms in spatially-varying coefficient Poisson as shown in Section 4

**Boundary term** $g$**:** We define the boundary term using the parametric equation $g(x) = \sin(\pi\Phi_\alpha x_0)\cos(2\pi\Phi_\alpha x_1) + (1 - \cos(\pi\Phi_\alpha x_0))(1 - \sin(2\pi\Phi_\alpha x_1)) + \sin^2(3\pi\Phi_\alpha x_2)$, where $\Phi_\alpha$ is the diffusion frequency. We denote that $u(x) = g(x)$.

**Diffusion Coefficient** $\alpha$**:** Diffusion coefficient describes the rate of diffusion of a physical quantity (e.g. heat) in a spatial medium. We define it as $\alpha(x) = \exp(-x_1^2 + \cos(4\pi\Phi_\alpha x_0)\sin(3\pi\Phi_\alpha x_1))$

**Absorption Coefficient** $\sigma$**:** Absorption coefficient allows for damping and amplification and is denoted as $\sigma(x) = A_{min} + (A_{max} - A_{min})(1 + 0.5\sin(2\pi x_0)\cos(0.5\pi x_1))$, where $A_{max}$ and $A_{min}$ are coefficients representing the minimum and maximum values for absorption.

**Source term** $f$**:** We compute the source term as $f(x) = -\alpha\nabla^2 u - \nabla\alpha \cdot u + \sigma u$. We approximate the gradient and the Laplacian from the closed-form expression.

## D    EXPERIMENTAL SETTINGS

### D.1    DOMAIN GENERATION

**Linear Setting**    For the linear settings, we generate the domain $\Omega_T$ by sampling points in the polar coordinate system. The domain $\Omega_T$ is disc-like, centered at the origin with its boundary radius defined by $r(\theta) = r_0[1 + c_1\cos(4\theta) + c_2\cos(8\theta)]$. We fix $r_0 = 1$, and $c_1, c_2$ are varying parameters uniformly sampled such that $c_1, c_2 \sim \mathcal{U}(-0.2, 0.2)$. To sample points inside the domain, we use a rejection-sampling strategy to ensure that all points lie strictly inside the boundary. For any candidate point $(x_0, x_1)$, we compute its polar coordinates, specifically the angle $\theta = \arctan2(x_1, x_0)$ and the radial distance $r = \sqrt{x_0^2 + x_1^2}$. A point is deemed to belong to the domain if $r < r_0$, where $r_0$ is calculated for the given $\theta$. We first over-sample a uniform grid of candidate points in the rectangular region enclosing $\Omega$. Each candidate point is evaluated against the domain boundary to determine whether it is within. Points that satisfy this criterion are retained, and the desired number of domain points is then sampled uniformly from the retained set. This approach ensures that the sampled points represent the geometry of the domain. In contrast, boundary points are explicitly generated by evaluating parametric boundary equations for uniformly spaced values of $\theta$. This guarantees an accurate representation of the domain boundary for enforcing Dirichlet boundary conditions.

**Varying Coefficient Setting**    For the experiments with Poisson with varying coefficients, we use meshes from the ShapeNet dataset (Chang et al., 2015) to define complex geometries. To sample domain and boundary points, we compute a fast signed distance field (SDF) estimate using the Bounding Volume Hierarchy (BVH). Domain points are identified by evaluating the SDF of a set of candidate points relative to the mesh, where points with a negative SDF (indicating that they lie inside the mesh) are retained. Boundary points are sampled by selecting points with an absolute SDF value within a small threshold $\epsilon = 0.01$, corresponding to points located at a distance of approximately $\epsilon$ from the mesh surface. The use of BVH speeds up the SDF computation by efficiently finding the nearest points on the mesh surface.

---

**Algorithm 1:** Training of Vanilla WoS-NO without Caching

---

1 **Input:** Neural operator $\mathcal{G}_\theta$ parameterized by $\theta$, the number of epochs $E$, distance function $T$, source term $f$, bound term $g$, Walk-on-Spheres operator $\hat{\mathcal{G}}_{L,\text{WoS}}$, the number of trajectories $L$, domain $\Omega_T$, input function space $\mathcal{A} = \mathcal{T} \times \mathcal{F} \times \mathcal{B}$, learning rate $\gamma$

2 **Output:** Learned parameters $\theta$

3 **for** $i \leftarrow 0...E$ **do**

4 $\quad$ $a = (T, f, g) \leftarrow$ sample from $\mathcal{A}$

5 $\quad$ $\{\xi^j\}_j \leftarrow$ sample from $\Omega_T$

6 $\quad$ $\{y^j\}_j \leftarrow \text{vmap}[\hat{\mathcal{G}}_{L,\text{WoS}}[a](\{\xi^j\}_j)]$

7 $\quad$ $\hat{\mathcal{L}}(\theta) = MSE(\mathcal{G}_\theta[a](\xi^j), y^j)$

8 $\quad$ $\theta \leftarrow \text{step}(\gamma, \nabla_\theta\hat{\mathcal{L}}(\theta))$

9 **end**

---

## E    TRAINING DETAILS

We train our model on a set of 4000 PDE instances defined over unique geometries. As our framework is primarily architecture independent, we leverage 3 architectures within the neural operator class -

GINO Li et al. (2024b), GNOT Hao et al. (2023) and Transolver Wu et al. (2024). During training, we sample 1024 domain and boundary points, respectively, from the instance and this forms the input to the model, alongside the pointwise source, (diffusion, and absorption for the varying coefficient case) values computed on the samples. We evaluate the performance on unseen geometries and unseen PDE parameters. To train the model, we use the Adam optimizer, with a learning rate of 1e-3 and a weight decay of 1e-6. We use the linear WoS-based residual for the linear and the delta tracking residual for the varying coefficient one. The training process includes a learning rate scheduler (*ReduceLROnPlateau*), which reduces the learning rate by a factor of 0.9 when the validation loss stagnates for 2 consecutive epochs. All experiments were performed on NVIDIA RTX 3060 GPU.

### E.1 HYPERPARAMETERS FOR THE MODELS

**GINO** For the GINO module, an embedding dimension of 16, with a maximum of 600 positional embeddings. The graph construction radii for input, and output, were set to 0.1, 0.1, and 0.05, respectively. The input and output transforms are linear; the input channel MLP has hidden layers of [4, 256, 512, 256, 3] and the output channel MLP has layers of [128, 512, 1024, 512, 64]. The FNO module consists of 4 layers with [20, 20] Fourier modes. The hidden, lifting, and projection channel dimensions were set to 64, 256, and 64, respectively. We used Group Normalization, a channel MLP expansion factor of 0.5, 8 features for AdaIn, and a tensor factorization rank of 0.8.

**GNOT** The GNOT model is a Transformer-based architecture with 12 layers with a hidden dimension of 128. The attention mechanism employs a single head with a linear attention type and no dropout. The feed-forward network is a Mixture-of-Experts model with 2 experts. Each expert's MLP is composed of 2 layers with an inner dimension of 8, and no dropout is applied. Horizontal Fourier features were not used.

**Transolver** The Transolver model is a Transformer-based architecture configured with 6 layers, a hidden dimension of 32, and 24 attention heads. The model incorporates residual connections and is set with a slice number of 32 and a reference value of 16. It does not use time as an input feature and is designed to produce a single scalar output.

## F ABLATIONS

In this section, we provide different ablation studies for improving the performance of Wos-NO.

### F.1 EFFECTS OF CONTROL VARIATES

As discussed in (Sawhney & Crane, 2020), control variates provide a principled approach to reducing the variance of Walk-on-Spheres (WoS) estimates, leading to more consistent and accurate results. However, in our case, we observe no significant improvements, due to the expressivity of neural operators and their ability to learn from noisy inputs.

Table 3: Showcasing the effects of control variates on the 2D dataset.

| CONTROL VARIATES | TRAIN L2 Err | TEST L2 Err |
|---|---|---|
| No | $4.05e^{-4}$ | $8.2e^{-4}$ |
| Yes | $4.33e^{-4}$ | $8.35e^{-4}$ |

### F.2 CACHING PRIOR WALKS

Inspired by (Nam et al., 2024), we use a cache to store intermediate walks $\{(a^j, \xi^i, \hat{\mathcal{G}}_{L,\text{WoS}}[a^j](\xi^i))\}_{i,j}$. Instead of relying on a fixed, noisy target, the cached estimate is updated each epoch with a small number of fresh Monte Carlo walks. This process amortizes the variance reduction over the training duration. The cache incurs an additional space complexity of $O(mn)$, where m is the number of instances in the training dataset and n is the number of points per instance.

Let k be the current training epoch and L be the number of new WoS trajectories generated per epoch for each point. The cached target estimate at epoch k, which we denote as $Y^{(k)}$, is the running average of all $k \cdot L$ trajectories generated up to that point. This can be expressed via the following

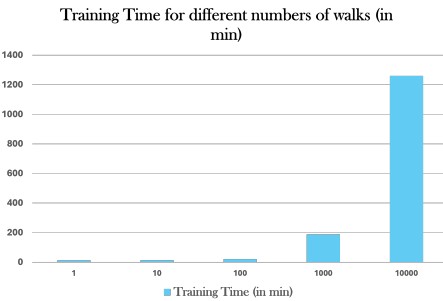 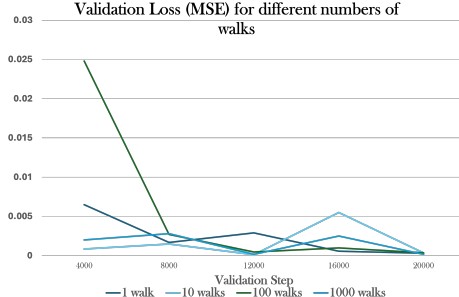

Figure 7: **Left:** This figure showcases the overall training time as the number of trajectories changes for each instance. The higher number of trajectories increases the training time due to a higher likelihood of longer random walks. **Right:** This plot shows comparable performance for different numbers of walks for each instance. Asymptotically, neural operators converge to a similar level of errors regardless of the fidelity of the stochastic estimation.

recursive update rule:

$$Y_{j,i}^{(k)} = \frac{k-1}{k} Y_{j,i}^{(k-1)} + \frac{1}{k} \hat{\mathcal{G}}_{L,\text{WoS}}^{(k,\text{new})}[a^j](\xi^i) \quad \text{with} \quad Y_{j,i}^{(0)} = 0 \tag{32}$$

where $\hat{\mathcal{G}}_{L,\text{WoS}}^{(k,\text{new})}$ is the estimate derived from the $L$ fresh walks generated during epoch $k$. As the training progresses ($k \to \infty$), the variance of our target approaches zero, and $Y_{j,i}^{(k)}$ converges to the true solution $\mathcal{G}[a^j](\xi^i)$. The empirical loss at epoch $k$ is therefore a function of this refined target:

$$\hat{\mathcal{L}}_\theta^{(k)} = \frac{1}{NM} \sum_{j=1}^{M} \sum_{i=1}^{N} \|\mathcal{G}_\theta[a^j](\xi^i) - Y_{j,i}^{(k)}\|^2 \tag{33}$$

### F.3 EFFECTS OF NUMBER OF TRAJECTORIES

To evaluate the impact of the number of trajectories for WoS-NO. We train WoS-NO with $L = 1, 10, 100, 1000$ to evaluate the influence of the number of trajectories for WoS-No.

As shown in Figure 7 left, the number of trajectories has a direct influence on the training time. The higher the number of trajectories, the lower the variance and the longer the simulation time for full convergence of random walks. This results in a trade-off between fidelity and simulation time and requires careful tuning to balance two costs. We further evaluated the validation loss for different numbers of trajectories in Figure 7 right. We discover that asymptotically, the performance of WoS-NO converges to a similar validation loss level, proving that neural operators are able to smooth out noises in stochastic estimations in the long term. We choose $L = 10$ as we find that it has slight improvements in the final performance while not introducing significant costs for training.

