# OpenReview forum: "OPERATOR LEARNING USING WEAK SUPERVISION FROM WALK-ON-SPHERES"
_ICLR.cc/2026/Conference — Submitted to ICLR 2026_

### Official Review · Reviewer_f4PC · 2025-10-23

**Soundness:** 3
**Presentation:** 3
**Contribution:** 2
**Rating:** 4
**Confidence:** 4

**Summary:**

This paper present an interesting training methodology for neural operators. They make use of the walk-on-sphere Monte Carlo algorithm to generate noisy, unbiased, estimated of Poisson problems and regress a NO to it's mean. They focus on the standard linear Poisson problem and a modified version of a Poisson with variable coefficients. The method is architecture agnostic.

**Strengths:**

- The paper is well written, and in the majority, easy to follow.
- The paper presents a creative use of the WoS algorithm which seems to work well with learning surrogate models.
- The paper explores a few interesting applications. In particular, the inpainting example is creative.
- The experiments are well conducted and report relevant metrics, if however limited in terms of benchmarks against standard numerical solvers.

**Weaknesses:**

- The paper claims to apply their methodology to a **nonlinear** Poisson PDE. The equation presented is linear in the solution with variable coefficients. This is a significant limitation.

- Overall the Poisson PDE is of some practical use, however it is generally used a 'fruit-fly' type problem for PDEs, particularly in the context of neural operators as direct solvers are already very efficient for such problems. Neural operator mainly find their use in nonlinear problems. The method being specific to linear Poisson-type PDEs greatly limits its practical use.

- WoS is not the standard method for solving Poisson PDEs. The authors do not use classical solvers such a conjugate-gradients to benchmark their method.

- This is a minor point, but the mathematics in some places is overly terse. I would suggest added a few sentences to help readers less familiar with stopping time MC estimates of harmonic functions and WoS.

**Questions:**

- The first 4 short paragraphs in section 3 are extremely terse. Can these be expanded (or justified why this is necessary) to help the reader understand why you are taking care in being specific. For example, what is the purpose of invoking $D$, $\mathcal{T}$, $S_T$, etc to then only use $\Omega$? Could we have equivalently assumed $\Omega$ to be open and Lipschitz?

- In figure 4, I cannot see the runtime for WoS-NO.

---

> ### Author Response · Authors · 2025-11-22
>
> For responses to W1-W3, we would like the reviewer to refer to the general responses we made
>
> **The mathematics in some places is overly terse**
>
> Thank you for the suggestion! We have made significant simplifications of our problem setup sections so that it is more accessible to a general audience while not losing the full mathematical rigor.
>
> **The first 4 short paragraphs in section 3 are extremely terse.**
>
> Thank you for pointing this out! We tried to be as formal as possible in the beginning to ensure we are not missing any mathematical detail, but we definitely agree that open and Lipschitz domain assumption should be sufficient. We made significant simplifications in our updated PDF to be more accessible to the general audience while not losing technical details. Please refer to the updated draft!
>
> **I cannot see the runtime for WoS-NO.**
>
> Thank you for pointing this out! GINO in the figure is the same as WoS-NO, as proven to show the strongest performance in comparison with Transolver and UPT. We have changed the figure to be more explicit.

---

> > ### Comment · Reviewer_f4PC · 2025-11-24
> > **Response to Rebuttal**
> >
> > I thank the authors for their careful consideration of my, and the other reviewers, comments. In particular, addressing the claim that the method was used on nonlinear PDEs was important. Overall I believe this to be a very original contribution and well executed contribution.
> >
> > I will request one last thing from the authors: Can you expand on the extensions of your work to the other PDEs you mention (Transient diffusion problem, Reaction-Diffusion equation, Lamé Equation, Electron transport equation, Navier-Stokes, and Burgers equations). If you could include a paragraph clearly outlining how such extensions may be possible, this is of very high importance for the practical relevance of the work.
> >
> > I have updated my score to reflect the changes you have made.

---

> > > ### Author Response · Authors · 2025-12-03
> > >
> > > We would like to thank the reviewer for raising the score to 6!
> > >
> > > Our framework only requires that the PDE solution can be represented as an expectation of a suitable path functional. Let $D$ denote either a spatial domain $\Omega \subset \mathbb{R}^d$ or a space–time cylinder $D = \Omega \times [0,T]$. Consider a linear parabolic/elliptic problem of the form
> > >
> > > $$
> > > \partial_t u(t,x) + \mathcal{L} u(t,x) + c(t,x)u(t,x) = f(t,x),
> > > \qquad (t,x) \in D,
> > > $$
> > >
> > > with terminal and/or boundary conditions, where
> > >
> > > $$
> > > \mathcal{L}u(t,x)=\sum_{i,j=1}^d a_{ij}(t,x)\partial_{x_i x_j}^2 u(t,x)+\sum_{i=1}^d b_i(t,x)\partial_{x_i} u(t,x)
> > > $$
> > >
> > > is a (possibly anisotropic) second-order operator, and $c$ is a bounded reaction (or potential) term. Under standard assumptions (uniform ellipticity, regularity of coefficients), the solution admits a Feynman–Kac representation in terms of a diffusion process $(X_s)_{s \ge t}$ with generator $\mathcal{L}$,
> > >
> > > $$
> > > \mathrm{d}X_s=b(s,X_s)\mathrm{d}s+
> > > \sigma(s,X_s)\mathrm{d}W_s,
> > > \qquad
> > > \sigma \sigma^\top = 2A = (2a_{ij})_{i,j}.
> > > $$
> > >
> > > Let $\tau$ be the relevant stopping time (exit time from $\Omega$, terminal time $T$, or a combination). For appropriate boundary/terminal data $g,\varphi$, one has
> > > $$
> > > u(t,x)=
> > > \mathbb{E}_{t,x}\Bigg[ \exp\Big(\int_t^{\tau} c(s,X_s)\mathrm{d}s\Big) \Phi\Big( X _\tau \Big) + \int_t^{\tau} \exp\Big(\int_t^{s} c(r,X _r)\mathrm{d}r\Big)f(s,X _s)\mathrm{d}s \Bigg],
> > > \tag{*}
> > > $$
> > > where $\Phi$ encodes the boundary/terminal condition. In particular, the solution $u$ is an expectation of a path functional of the Markov process $X$.
> > >
> > > Using the tower property (as in Section 3.2, Equation (5)), we define our ideal loss as
> > >
> > > $$
> > > \mathcal{L}(\theta)=\mathbb{E}\big[\|\mathcal{G} - \mathcal{G}_\theta \|^2\big]=\mathbb{E} _{a,\xi}\big[\big|\mathcal{G}\[a\](\xi) - \mathcal{G} _\theta\[a\](\xi)\big|^2\big],
> > > $$
> > >
> > > where $\mathcal{G}$ is the true solution operator, $\mathcal{G}_\theta$ is the learned neural operator, $a$ is the input coefficient field, and $\xi$ is the query point. Equivalently, by conditioning on $(a,\xi)$,
> > >
> > > $$
> > > \mathcal{L}(\theta)=\mathbb{E}_{a,\xi} \Big[\mathbb{E}\big[\big|\mathcal{G}\[a\](\xi) - \mathcal{G} _\theta\[a\](\xi)\big|^2\big|a,\xi\big]\Big].
> > > $$
> > >
> > > In practice, we cannot evaluate $\mathcal{G}\[a\](\xi)$ exactly, but we can approximate it using a Monte Carlo scheme (WoS, GRWG, transport MC, etc.) based on the representation $(*)$. Let $\widehat{\mathcal{G}}^{(l)}\[a\](\xi)$ denote the estimator from the $l$-th Monte Carlo path, and define the $L$-sample averaged estimator
> > >
> > > $$
> > > \widehat{\mathcal{G}}^{(L)}\[a\](\xi):=\frac{1}{L}\sum_{l=1}^L \widehat{\mathcal{G}}^{(l)}\[a\](\xi).
> > > $$
> > >
> > > By construction,
> > >
> > > $$
> > > \mathbb{E}\big[\widehat{\mathcal{G}}^{(L)}\[a\](\xi)\big]=\mathcal{G}\[a\](\xi),\qquad \mathrm{Var}\big(\widehat{\mathcal{G}}^{(L)}\[a\](\xi)\big) = \mathcal{O}\left(\tfrac{1}{L}\right),
> > > $$
> > >
> > > so $\widehat{\mathcal{G}}^{(L)}$ is an unbiased estimator with the usual Monte Carlo convergence rate $L^{-1/2}$ in standard deviation. We therefore approximate the loss by
> > >
> > > $$
> > > \mathcal{L}(\theta)\approx\mathbb{E}_{a,\xi}\big[ \big| \widehat{\mathcal{G}}^{(L)}\[a\](\xi)-\mathcal{G} _\theta\[a\](\xi)\big|^2\big].
> > > $$
> > >
> > > This construction does not depend on the PDE being Poisson; it only uses the existence of a representation of the form $(*)$ and an associated unbiased Monte Carlo estimator.

---

> ### Author Response · Authors · 2025-12-03
>
> At a high level, the stochastic solvers in [2–6] share two common design principles that align naturally with our framework:
>
> 1. **Reduction to linear problems with Feynman–Kac representations.**
>
>    Nonlinear or coupled PDEs (such as reaction–diffusion systems, Lamé elasticity, semiconductor transport, or Navier–Stokes/Burgers equations) are rewritten as sequences of linear parabolic/elliptic problems, often via fixed-point or Picard-type iteration. Each linear subproblem has a generator $\mathcal{L}^{(m)}$ and admits a representation of the form $(*)$ for some process $X^{(m)}$. In other words, even when the original PDE is nonlinear, the numerical scheme works by repeatedly solving linear problems that fit the Feynman–Kac template.
>
> 2. **Random-walk approximation of exit/terminal distributions.**
>
>    For each linear subproblem, the underlying Markov process is simulated via a random-walk algorithm that approximates the distribution of $(X^{(m)}_s) _{s \le \tau^{(m)}}$ and its exit/terminal time $\tau^{(m)}$. Examples include:
>    - Walk on Spheres (WoS): steps are taken to the boundary of the largest inscribed sphere around the current point, with the exit point drawn from the (possibly anisotropic) exit distribution and the corresponding exit time sampled or approximated;
>    - Global random walk on grid (GRWG): steps are taken on a regular grid with transition probabilities tailored to the discretized operator $\mathcal{L}^{(m)}$;
>    - Transport Monte Carlo: trajectories in phase space (e.g., for Boltzmann transport) are simulated with stochastic scattering events and pathwise weights.
>
> In all these cases, the outcome of the stochastic solver at a point $\xi$ is precisely a path-functional estimator $\widehat{\mathcal{G}}^{(L)}\[a\](\xi)$ of the form induced by $(*)$. Our contribution is to use such estimators not as an end goal, but as weak supervision for a neural operator: we amortize the cost of Monte Carlo across many problem instances and queries by learning $\mathcal{G}_\theta$, while relying on the existing stochastic methods solely to supply unbiased (or controlled-bias) training signals.

---

### Official Review · Reviewer_t1DZ · 2025-10-28

**Soundness:** 2
**Presentation:** 2
**Contribution:** 2
**Rating:** 4
**Confidence:** 2

**Summary:**

This paper proposes an innovative neural operator training framework called WoS-NO, which utilizes weak supervision signals generated by the Walk-on-Spheres (WoS) stochastic process to train neural operators. This approach avoids the expensive pre-computed data generation and unstable Physics-Informed Neural Network (PINN) optimization traditionally associated with solving Partial Differential Equations (PDEs). The core idea is to use unbiased estimates from a minimal number of WoS trajectories as training targets, enabling the neural operator to learn to "denoise" and converge to the true solution, thereby achieving data-free and derivative-free operator learning.

**Strengths:**

1. **Well-Defined and Significant Problem:** The paper accurately identifies a critical bottleneck in current PDE solving: the high cost of pre-computed data and the instability of PINN optimization. Addressing this issue is crucial for advancing practical applications in scientific computing.
2. **Novelty of the Weakly-Supervised Training Strategy:** The proposed idea of using the WoS stochastic process for weak supervision is pioneering in this field. It cleverly combines the unbiased nature of Monte Carlo methods with the representational power of neural networks, offering a fresh perspective on solving PDEs.

**Weaknesses:**

1. **Limited Scope of Problem Types:** The method is primarily applied to the family of Poisson equations. Its applicability to a broader range of PDE types (e.g., Navier-Stokes equations, wave equations) remains unverified. The paper should discuss the potential and challenges of extending this framework to other important PDE classes.
2. **Lack of Systematic Study on WoS Parameter Selection:** The choice of the number of WoS trajectories (L ≤ 10) appears empirical. There is a lack of systematic analysis on how different L values affect training efficacy and final performance. An ablation study on the hyperparameter L is recommended.
3. **Insufficient Integration with Modern Variance Reduction Techniques:** While WoS estimates are unbiased, they inherently possess high variance. The paper would benefit from exploring the integration with other advanced variance reduction techniques (e.g., importance sampling, stratified sampling) to control this variance and improve stability.
4. **Insufficient Validation on Large-Scale Problem Scalability:** Experiments are conducted primarily on medium-scale problems. The scalability of the method to very high-dimensional or large-scale industrial-level problems requires further demonstration.

**Questions:**

See Weaknesses

---

> ### Author Response · Authors · 2025-11-22
>
> For responses to W1, we would like the reviewer to refer to the general response.
>
> **Lack of Systematic Study on WoS Parameter Selection**
>
> Thank you for pointing this out! To make a fair comparison, we ran the experiment by training GINO with WoS trajectories in the range of 1, 10, 100, and 1000 to understand the impact of the solution. The number of trajectories can be considered as a parameter to adjust the “strength” of the weak supervision, where more trajectories imply less noisy results while requiring more simulations. We show that while 1 trajectory is the fastest for convergence, its performance is not strong. We believe that this is because the supervision is way too weak and the noise dominated the solution. In contrast, after L >= 10, we show negligible improvements in generalization, suggesting that L = 10 is sufficient to achieve a strong performance. We added the new ablation study in Appendix F.
>
> **Insufficient Integration with Modern Variance Reduction Techniques**
>
> Thank you for pointing this out. As the reviewer has pointed out, it is critical to introduce variance reduction methods for Monte-Carlo simulations. Our WoS solver is based on the Zombie solver [1], and it already contains multiple techniques for variance reduction, such as control variates, Russian roulette, and importance sampling. We will consider other types of techniques as suggested by the reviewer in our future work to achieve stronger performance.
>
> **Insufficient Validation on Large-Scale Problem Scalability**
>
> Thank you for pointing it out. We would like to ask the reviewer what they consider a large-scale problem. We believe that running Poisson equations defined on a complex geometry is already a challenging task. As far as we are concerned, we are the first to amortize over different, complex shapes, enabling a one-shot solution for geometries where FEM solvers struggle without careful meshing. We tested the FEM solver on 3D ShapeNet, and it required significant tuning of mesh resolutions to make it even runnable. In contrast, WoS-NO requires no meshing and is easily runnable with strong performance over complex geometries, as shown in Figure 2. We believe that this already suggests that WoS-NO is a strong candidate that is scalable to large-scale problems with highly complex geometries. To further emphasize our advantage, please also refer to the general response on **Why WoS**.

---

### Official Review · Reviewer_2Nx6 · 2025-10-29

**Soundness:** 2
**Presentation:** 2
**Contribution:** 2
**Rating:** 2
**Confidence:** 3

**Summary:**

The proposed paper studies solving Poisson PDE using stochastic Walk-on-sphere methods. In a first time, the method is introduced and the Walk-On-Sphere algorithm explained. Finally, the method is evaluated against a selection of baslines on several tasks.

**Strengths:**

- The experimental results showcase an improvement wrt to baseline.
- The proposed method is both GPU-memory and time efficient and performs best on the studied problems.

**Weaknesses:**

-	My main concerns is about the scope of applicability : in my understanding the proposed method applies only to a limited familly of PDE : Poisson PDE.
-	It is not detailed why focusing that much on such PDEs is important.
-	The paper is hard to read, I think some re-writing would help the understanding of the paper (eg in section 2.2 which give a lot of references which makes the paragraph hard to follow).  I felt hard to understand the key objectives of the paper.
-	Some experimental details/settings are missing, making the paper hard to position and understand.
-	No reference to some figures in the text (eg figure 2)

**Questions:**

### Questions :
-	Could you illustrate/detail about the O(1) stated in the introduction line 107. Is it illustrated in the experiments ?
-	Is the method limited to Poisosn PDEs ? What application could it be used for ? Why focusing that much on such PDEs? Is the method applicable on other PDEs? I saw an experiment on a Laplace PDE, but is it applicable to other PDE/datasets, that are commonly used in the PDE community?
-	Could you detail the PDEs setting ? What are considered in « new PDE instances » ? OOD or In-distribution evaluations ?
-	It is hard to make a link with the proposed figures, particularly figures 1 and 2.
-	In figure 3, why are the curves stopping before 200 mins ? I guess this is because the training time is lower, but I can’t find any justification on that point
-	Could you detail what are PINO losses ? A combination of a MSE loss with a residual pinns loss ?
-	Figure 4, right, what is your run time for comparison?
-	Figure 5 : how do we know/compare the proposed visualization with ground truth ?
-	Why providing wandb screenshots (I guess) of the GPU utilization? These figures are hard to read, I think such figure could be summarized in a table.

### Minor comments :
-	I think bolding the best metrics in tables helps reading the experimental parts.

---

> ### Author Response · Authors · 2025-11-22
>
> For extensive analysis on some of the weaknesses and questions, please also refer to our general responses.
>
> **It is not detailed why focusing that much on such PDEs is important.**
>
> We thank the reviewer for their concern about the focus of our paper. Poisson PDE is quite foundational in multiple areas in scientific computing, including potential theory, electrostatics, gravity, incompressible flow pressure, screening Poisson variants, fractional Poisson (anomalous transport), etc. We believe that having an efficient zero-shot solver for Poisson equations will definitely help the community perform high-fidelity simulations more efficiently.
>
> **The paper is hard to read,**
>
> Thank you for your suggestion. We have definitely rewritten those sections in place to make it more readable for readers. Here is a brief list of requested changes for improved readability.
>
> 1. We removed unnecessary references.
>
> 2. We significantly simplified the mathematics in the background for general accessibility.
>
> 3. We clarified our problem scopes in Section 1 for understanding the key objectives of our paper.
>
> 4. We added more experiments and comparisons with FEM solvers.
>
> The key objective of the paper is to show that, provided a certain systematic methodology of stochastic solvers, we can approximate the ground truth with stochastic simulation and use those noisy but unbiased estimations as our solutions to train neural operators instead of using PINN loss. We want to illustrate that such a weak supervision paradigm saves GPU memory (shown in Table 1, 2) with stronger performance.
>
> **Some experimental details/settings are missing**
>
> Thank you for pointing this out. Appendices D and E contain experiment details as well as training setups. We kindly ask the reviewer which details were missing, and we will add all those details for better reproducibility.
>
> **No reference to some figures in the text**
>
> Thank you for pointing out the issue! We have added references to Figure 2 in the text!
>
> **Could you illustrate/detail about the O(1) stated in the introduction line**
>
> Thank you for pointing this out. We stated O(1) as a point to show that, provided a well-trained neural operator, we can directly evaluate over arbitrary query points in parallel instead of requiring numerical solvers (e.g. FEM), which needs to do meshing + iterative solve. This amortization is illustrated in Figure 6, newly added to the paper. We empirically demonstrated that WoS-NO does not incur extra computation cost for inference with an increase in resolutions, in contrast to FEMs.
>
> **Could you detail the PDEs setting? What are considered in new PDE instances**
>
> The detailed PDE setting for 2D and 3D cases is shown in Appendix D. A new PDE instance is considered a random input function sampled from the PDE family. Probabilistically, it is measure zero to resample the same input function from the function family, so we are always testing over new instances that have never been seen during training.
>
> **It is hard to make a link with the proposed figures, particularly figures 1 and 2**
>
> Thank you for asking! We will definitely try to make it clearer to understand the link. Figure 1 is an overview diagram illustrating the pipeline of WoS-NO. Figure 2 is an illustration of the performance in 3D Poisson equation for zero-shot inference comparing WoS-NO, PINO, DeepRitz and WoS. We show that WoS-NO achieves better relative absolute error in different complicated shapes in comparison with all baselines. Darker means worse (larger error) performance
>
> **In figure 3, why are the curves stopping before 200 mins? I guess this is because the training time is lower, but I can’t find any justification on that point**
>
> Thank you for asking! We marked the stopping time to be sufficient for most methods to converge, and it was 200 minutes. PINO takes a lot longer to achieve the same performance. To make a fair comparison, we restricted the time to when most methods except PINO have converged. This is similar to the experimental settings in [1,2].
>
> **Figure 4, right, what is your run time for comparison?**
>
> Thank you for pointing out a typo here. We mistakenly marked WoS-NO as GINO (which is the backbone architecture of WoS-NO). The run time is 2.8 seconds for 20 masks. Significantly faster than using WoS solver directly (557.4 seconds) and scikit-image (6.8 seconds).
>
> ---
>
> [1] Sawhney, Rohan, et al. "Grid-free Monte Carlo for PDEs with spatially varying coefficients." ACM Transactions on Graphics (TOG) 41.4 (2022): 1-17.
>
> [2] Li, Zilu, et al. "Neural caches for monte carlo partial differential equation solvers." SIGGRAPH Asia 2023 Conference Papers. 2023.

---

> ### Author Response · Authors · 2025-11-22
>
> **Figure 5 : how do we know/compare the proposed visualization with ground truth**
>
> We used the baseline fluid simulation model [1], which uses the WoS solver for the projection step. We replaced the WoS solver with pre-trained WoS-NO for zero-shot inference on the solution for the projection. The final simulation results show the relative error of 2.5 x 10^-1 with WoS-NO, proving its generalization ability to unseen Poisson equations with high efficiency. In Figure 5, the top row indicates the ground truth, while the bottom row illustrates the simulated solution with WoS-NO. We compared the time for the WoS solver and WoS-NO and illustrated that WoS-NO achieves impressive generalization.
>
> **Why providing wandb screenshots (I guess) of the GPU utilization? These figures are hard to read, I think such figure could be summarized in a table.**
>
> We agree that, indeed, wandb screenshots might be less intuitive. We have removed them and added their information in tables (peak GPU memory usage) and plots (convergence time).
>
> **I think bolding the best metrics in tables helps reading the experimental parts.**
>
> Thank you for the suggestion! We bolded the best-performing model with the lowest error in Tables 1 and 2 to make it easier to read.
>
> ---
>
> [1] Jain, Pranav, et al. "Neural Monte Carlo Fluid Simulation." ACM SIGGRAPH 2024 Conference Papers. 2024.

---

> > ### Comment · Reviewer_2Nx6 · 2025-11-28
> > **Answer to author Rebuttal**
> >
> > I thank a lot the authors for their careful answers to my questions, as well as their answers to other reviewers' concerns. They have updated the paper and specifically the have improved the introduction and contextualization of their work.
> >
> > - Regarding W4: some of the experimental details I was referring to, were asked in the questions and you already answered. I would have an additional question regarding the PDE setup: could you detail what are considered in or out-of-distribution ? For this point, I refer to the zero-shot generalization claim for which I think further details would help. Are new parameters inside the training distribution PDE parameters ?
> > - In the newly added Figure 6, could you specify what does training refers to for the FEM baseline? The resolution time?
> >
> > I thank again the authors for answering my questions and updating the paper accordingly. I have raised my score accordingly.

---

> > > ### Author Response · Authors · 2025-12-03
> > >
> > > Thank you for raising the score! We will further clarify these two points:
> > >
> > > 1. In traditional sense, we tested both in and out-of-distribution settings. For table 1 and 2, we defined a fixed family of PDE families and trained and tested on new instances sampled from the same distribution (one can define that both cases are always seeing only new instances, so we are not cheating by showing test sets during validation). For Laplace inpainting and fluid dynamics, we used our operator in a completely new PDE family, so we tested the performance out-of-distribution. As shown by its low errors, we can claim that WoS-NO indeed learns a general solution operator.
> > > 2. Training time represents the time required for the neural operator to learn the data-driven objective from the points sampled from the FEM solution. Resolution means resolution of the input grid, and higher the resolution means finer details (inversely proportional).

---

### Official Review · Reviewer_ysKH · 2025-10-31

**Soundness:** 3
**Presentation:** 3
**Contribution:** 2
**Rating:** 6
**Confidence:** 3

**Summary:**

The paper introduces a weakly supervised learning framework for solving Poisson equations, where training labels are generated using the Walk-on-Spheres (WoS) algorithm. Unlike prior works that integrate WoS directly into neural architectures to simulate stochastic trajectories, this approach leverages WoS purely as a label generator, producing inexpensive yet noisy approximate solutions for a large number of Poisson PDE instances. A neural operator is then trained to regress toward these approximate solutions, effectively learning to denoise and generalize beyond. Once trained, the resulting operator demonstrates strong generalization ability to new Poisson problems with unseen boundary conditions and source terms, without requiring any additional fine-tuning.

**Strengths:**

- The paper demonstrates that inexpensive unbiased estimates of the solution can be effective for physics-informed neural operator learning. This aligns with the broader trend of scaling training data using pseudo-labels across wider problem domains.
- Some efforts have also been made to extend the framework to nonlinear PDEs.
- As the approach relies on pseudo-labeling, the proposed framework remains orthogonal to the choice of neural network architecture.

**Weaknesses:**

- Another related approach is PI-DeepONet (Wang et al., 2021), which adopts a PINN-style residual loss. It would be interesting to clarify how WoS-NO compares to this method, especially since PI-DeepONet is less constrained regarding the types of PDEs it can handle.
- Given the inherent specificity of the Walk-on-Spheres algorithm, WoS-NO appears applicable only to a limited class of PDEs. It would be valuable to discuss the potential for extending this framework to broader PDE families.

References:
- Wang et al. (2021), Learning the solution operator of parametric partial differential equations with physics-informed DeepONets

**Questions:**

- An analysis under varying numbers of training PDE instances would also be valuable for assessing data efficiency and scalability. In particular, it would be informative to examine how performance evolves as the number of sampled PDE instances increases—does accuracy continue to improve, or does it eventually saturate?
- It would be beneficial to include a baseline trained with high-fidelity PDE solutions to provide an orthogonal comparison with the current weakly supervised setting. Such an ablation would help clarify how the training regime—noisy versus accurate labels—affects both convergence behavior and generalization performance. For instance, to reach a comparable generalization error, how much accurate data versus noisy data is required?
- The paper could be further strengthened by emphasizing the unique advantages of the WoS algorithm, such as its natural scalability to high-dimensional Poisson problems and its potential to alleviate the restrictions on PDE forms typically faced by grid-based or deterministic solvers.

---

> ### Author Response · Authors · 2025-11-22
>
> Please also refer to our general responses, which answer some of the weaknesses and questions!
>
> **Impact of training instances on accuracy**
>
> During training, we are sampling new instances, or new samples of input functions, from the family of function classes and use WoS to generate coarse estimations that are used for training our neural operator. As shown in Figure 3 left, we observe that the accuracy saturates after training for a certain period and does not improve after a certain period. We believe that this implies the solution operator is discovered after a certain period. We tested over multiple backbone architectures, including GINO, Transolver and GNOT. As shown in Figure 3 right, we discovered that GINO achieves the lowest error.

---

### Author Response · Authors · 2025-11-22
**Common Questions**

**Paper Changes**

We have considered the feedback of reviewers to improve the presentation of our paper. To that end, we have made the following changes as requested. These changes are highlighted in blue in the paper.
1. Discussing the motivations for using Poisson problems (ysKH31, 2Nx630, t1DZ28, f4PC24)
2. Ablations on High fidelity solutions and WoS parameter tuning in Figure 7 (ysKH31, t1DZ28)
3. Cleaned up Literature Review and math explanations (2Nx630, f4PC24)
4. Extensions beyond Poisson in the introduction (2Nx630)
5. Clarifications on linearity (f4PC24)

**WoS-NO in comparison with PINN-style loss / What is PINO?**

We agree with the reviewer’s concerns on comparisons of WoS-NO with PINN-style residual loss. However, we want to point out several issues of PINN-based approach. PINO (Physics-Informed Neural Operator) [1] uses PINN-style loss and generalizes the PINN loss into operator learning, and the difference between PINN and PINO is demonstrated more in the PINO paper. To make a fair comparison, PINO is indeed one of our baselines. As shown in Table 1, PINO requires much larger GPU memory (2x),  in comparison with WoS-NO and requires longer training time (~7X) to achieve 10X worse performance for 2D problems. The key bottleneck with PINN approaches is the unstable gradient computation, which leads to training instabilities, and an increased training time. We had to spend quite some time tuning the hyperparameters to make the PINN training stable. This was much less of a case for WoS-NO, where the training was much more stable, with no unstable gradient computation.
As we check PINO in 3D problems, as shown in Table 2, the GPU memory scales much faster, making it difficult to scale PINO to more complicated higher-order PDEs.

**Why limit to Poisson families?**

We agree with reviewers that WoS-NO currently is applicable only to Poisson equations. However, there are many possible extensions of the framework. Currently, there exist several direct extensions of WoS to achieve
1. Transient diffusion problem [2]
2. Reaction-Diffusion equation [3]
3. Lamé Equation [4]
4. Electron transport equation [5]
5. Navier-Stokes and Burgers equations [6]

We also want to stress that using a linear family does not fully discredit our methodology. We believe that this is the first proof-of-concept to introduce weak supervision into neural operators. We discover that a lot of PDEs require intensive numerical simulations to estimate the solution over complex geometry. Our methodology verifies that, provided a methodology to make rough estimations of a family of PDE (e.g. WoS, FBSDE, BSDE, WoB), neural operators can learn from those noisy but unbiased estimators much more efficiently than pure PINN-style residual loss with much less GPU memory requirement.

At last, we want to stress that linear Poisson equations are still difficult to resolve with complex geometries. FEM requires significantly more precomputation time for discretizing the domain, and it is quite fragile to mesh discretization where bad discretization could easily make computation fail.

In summary, our key contributions are amortizing the training time by avoiding pre-computation of training data, avoiding the gradient-related instabilities of PINN, Linearization and reduction of problem settings to Poisson style formulation to facilitate WoS (i.e. Image inpainting, Pressure projection in fluid dynamics)

---

[1] Li, Zongyi, et al. "Physics-informed neural operator for learning partial differential equations." ACM/IMS Journal of Data Science 1.3 (2024): 1-27.

[2] Shalimova, Irina, and Karl Sabelfeld. "Random walk on spheres method for solving anisotropic transient diffusion problems and flux calculations." Monte Carlo Methods and Applications 30.1 (2024): 73-80.

[3] Sabelfeld, Karl. "Correlation Structure of the Solution to the Reaction-Diffusion Equation in Respond to Random Fluctuations of the Boundary Conditions." Fluctuation and Noise Letters 23.01 (2024): 2450001.

[4] Aksyuk, Ivan Alekseevich, et al. "Stochastic Simulation Algorithms for Iterative Solution of the Lamé Equation." Numerical Analysis and Applications 16.4 (2023): 299-316.

[5] Kablukova, Evgeniya, et al. "Stochastic simulation of electron transport in a strong electrical field in low-dimensional heterostructures." Monte Carlo Methods and Applications 29.4 (2023): 307-322.

[6] Sabelfeld, Karl K., and Oleg Bukhasheev. "Global random walk on grid algorithm for solving Navier–Stokes and Burgers equations." Monte Carlo Methods and Applications 28.4 (2022): 293-305.

---

### Author Response · Authors · 2025-11-22
**Common Questions**

**Comparison with traditional solvers**

Thank you for pointing this out! There are multiple reasons we did not compare WoS-NO with traditional solvers such as FEM. First, FEM is a mesh-based method that introduces discretization errors. As a result, using FEM as the ground truth already introduces biases. We tried running FEM simulations on our 3D Poisson with varying coefficients over ShapeNet geometries. Since FEM is a mesh-based method, we include the domain discretization step and FEM computation time as the total time for inference. The big
We do agree that if we only consider the computation time, FEM simulation is much faster than WoS or any other Monte-Carlo algorithms. However, we want to suggest that pre-computation of discretization meshes for FEM is a significant bottleneck. We present this in Figure 6.

**Why WoS**

We agree with the reviewer on suggesting this! WoS is clearly advantageous when we have very complex geometries. To further point out why WoS is more advantageous than FEM-based solvers, we would like to cite Section 6.4 of [1], which compares FEM solver with WoS. While the computation for the FEM solver is much less, the discretization cost for precomputation is much higher than FEM solvers.
FEM computes the true solution over the entire mesh. While WoS is in principle, slower than FEM due to the need to run several walks, we amortize this into the training setup by ONLY computing a small number of walks, generating a weak estimate, making it significantly faster than FEM on a single instance. However, at each epoch, a new set of random walks is performed, allowing the operator to reduce variance
To verify this, we ran additional ablations on the effect of the number of walks on validation accuracy, and present the results in Figures 6 and 7.

---

[1]. Sawhney, Rohan, et al. "Grid-free Monte Carlo for PDEs with spatially varying coefficients." ACM Transactions on Graphics (TOG) 41.4 (2022): 1-17.

---

### Author Response · Authors · 2025-12-03

Dear Area Chairs,

We thank you as well as the reviewers for their constructive engagement throughout the rebuttal process. Prior to the system rollback, **all active reviewers have raised their scores (Reviewer 2Nx6, Reviewer f4PC) to 6** as shown in the rebuttal comments. For an overview, we summarize the strengths commented by the reviewers as well as their concerns and our replies to each points below.

### Strengths
1. **Memory-efficient and highly parallel:** Indeed, Monte Carlo methods like WoS do not require a discretization of the domain and are highly parallel as each individual trajectory is independent of each other. Thus, WoS can be used on-the-fly during training (using weak-supervision, see below) and we do not need to rely on computationally expensive data from, e.g., FEM solvers. Also, compared to other physics-informed neural methods such as PINOs, our simple regression loss is significantly more efficient since we do not require any higher-order derivatives.
2. **Novel use of weak-supervision:** Many reviewers pointed out our novel use of coarse estimators from stochastic solvers as weak supervision for training high-fidelity PDE surrogates. WoS-NO can be viewed as paradigm shift, showing that high-fidelity data is not crucial to obtaining strong performance. In particular, if we have efficient coarse estimators that are noisy but unbiased, we can (theoretically) still converge to the optimal operator. In practice, we show that this provides highly efficient and accurate neural PDE solvers.

### Key Concerns

Below, we summarize the key concerns raised and our responses.

1. **Why do we prefer using weak supervision over PINN and regression over traditional solvers?** As shown in Table 1 and Table 2, both PINN and DeepRitz require much more GPU memory than WoS-NO since they both rely on the computation of higher-order derivatives. In particular, PINNs exhibit poor memory scalability to higher dimensions (2D 1523.5MB->3D 8587.9MB). Moreover, we compare processing times for computing solutions with traditional solvers, such as finite-element methods (FEM), in Figure 6. Overall, precomputing data with FEM methods and training the operator requires significantly more computational resources due to the requirement of fine discretizations to obtain sufficiently accurate on complex geometries. Moreover, the data produced by FEM solvers is typically biased, which provides a lower bound on the performance of the neural operator trained on such data. While the bias can be reduced by refining the discretization, this incurs significantly higher computational costs. On the other hand, WoS is a meshless method that is guaranteed to produce unbiased solutions (even for low-fidelity estimates), theoretically allowing the neural operator to converge to the optimal solution.
2. **Why do we focus on Poisson equations? What are possible extensions?** We would like to point out Poisson equations are extremely important PDEs that frequently appear in areas such as fluid dynamics, electrostatics, and computer graphics. Moreover, while Poisson equations are linear, they are still challenging to solve over complex geometries with traditional methods. However, we also want to emphasize that our general framework is not restricted to Poisson equations. Our key idea is to show that, for any family of PDEs, we can efficiently train neural operators to high-fidelity surrogates as long as we have access to coarse but unbiased estimators. For instance, we also mention extensions of WoS methods to more general equations and applicability of our framework to Transient diffusion problems, Reaction-Diffusion equations, Lamé equations, Electron transport equations, Navier-Stokes equations, and Burgers equations.

For detailed responses to all individual questions, please refer to our previous general responses to common questions as well as comments we made for each reviewer!

---

### Meta-Review · Program_Chairs · 2026-01-03

**Summary:**

This paper trains neural operators for Poisson PDEs using noisy, unbiased Walk-on-Spheres Monte Carlo estimates as weak supervision.

**Reviewer Concerns:**

Main concerns from reviewers include:

1. Reviewers question practical scope: currently limited to Poisson/linear (variable-coefficient ≠ nonlinear) and unclear extension beyond WoS.

2. They request stronger baselines (PI-DeepONet, classical solvers like CG/FEM), ablations (high-fidelity labels, #WoS walks L, variance reduction), clearer scalability/zero-shot OOD definitions, and improved writing/experimental details.

**Reviewer Scores:**

Review / Old Score / New Score

ysKH	6	6

2Nx6	2	4

t1DZ	4	4

f4PC	4	6

Average	4	5

---

2Nx6 and f4PC mentioned they would raise their scores, though they provided no evidence as to by how much.

---

### Decision · Program_Chairs · 2026-01-26

Reject